# Position: Reliable Reward Design Needs Both Humans and Foundation Models

## Abstract

Reward design is central to reinforcement learning (RL), determining whether an agent successfully solves a task or exploits a specification loophole. While foundation models (FMs) can leverage broad semantic knowledge to synthesize reward functions from task descriptions, fully autonomous design invites undetected mistakes and misalignment that may go unnoticed until deployment. Conversely, manual reward engineering by domain experts ensures alignment but is time-consuming and difficult to scale. In this position paper, we argue that the RL community must pivot toward reward design frameworks that integrate human judgment with the generative capabilities of FMs. Placing humans in a loop to verify and refine FM-generated reward functions produces transparent, auditable reward artifacts, leveraging both the scalability of automated synthesis and the normative grounding of human intent.

## 1. Introduction

Reinforcement learning (RL) has achieved remarkable success in domains with well-defined objectives, from strategic game playing (Silver et al., 2017) to protein folding (Jumper et al., 2021). However, as we apply RL to increasingly complex tasks, defining correct reward functions becomes a severe bottleneck (Dulac-Arnold et al., 2021). An RL agent is only as good as the reward it optimizes; if the reward is misspecified, the policy will likely fail. For example, a robot blindly following an under-specified reward may clean a room by throwing everything out the window. In such cases, the failure is not in the learning algorithm, but in the *specification gap*: the disconnect between abstract human intent (e.g., "drive safely") and the concrete, scalar-valued reward from which the agent learns. Bridging this gap remains difficult—even when armed with practical guidelines for reward design (Knox & MacGlashan, 2024), slight misspecifications can lead to *reward hacking*, allowing an agent to exploit loopholes to maximize reward while violating the designer's intent (Amodei et al., 2016; Skalse et al., 2022).

foundation models (FMs) (Bommasani et al., 2021)—primarily large language models (LLMs) and vision language models (VLMs)—offer a solution. With their vast semantic background knowledge and code generation capabilities, FMs act as semantic bridges, translating ambiguous natural language descriptions into executable reward logic (Cao et al., 2025). This effectively lowers the barrier to entry, allowing designers to express complex behavioural constraints without manually defining every mathematical term. However, this capability has sparked a trend toward fully autonomous, zero-shot reward generation that we believe is dangerous. The core issue is not merely that FMs make mistakes, but that fully autonomous loops create a closed validation circuit. They treat reward design purely as a translation task, mapping text to a reward specification, rather than a grounding task so that it aligns with physical reality. Without human verification, the system risks optimizing a hallucinated proxy—a reward function that looks syntactically correct but incentivizes physically impossible or unsafe behaviour. By removing the human from the loop, we risk automating the generation of these subtle bugs, creating agents that optimize flawed rewards with superhuman efficiency. In short: An FM is a powerful engine for reward generation, but it lacks a steering wheel for intent.

This position paper argues that **reliable reward design requires collaboration between humans and FMs**. Building on the foundation of human in the loop (HITL)-RL (Retzlaff et al., 2024), we claim that human involvement is not merely advantageous, but a *prerequisite* for reliable FM-based design. Specifically, we argue that the design of complex reward specifications requires three critical components: an *auditable artifact* (explicit reward code), a *normative anchor* (the human verifying intent), and an *iterative repair loop* (the process of FM-driven refinement).

To operationalize this argument, we focus on the use of FMs as architects for RL agents, explicitly distinguishing this from the inverse paradigm of using RL to align the models themselves. Within this scope, this paper makes two contributions. First, we introduce a **design matrix**

[1]Anonymous Institution, Anonymous City, Anonymous Region, Anonymous Country. Correspondence to: Anonymous Author <anon.email@domain.com>.

Preliminary work. Under review by the International Conference on Machine Learning (ICML). Do not distribute.

**taxonomy** that categorizes existing literature along the axes of human involvement and reward representation, revealing a clear safety gap in fully autonomous methods. Second, addressing this gap, we advocate for a collaborative **reward design framework**. This "fast-slow" architecture assigns the FM to the fast loop (handling high-volume syntax and logic generation) and the human to the slow loop (providing high-leverage semantic patches). We conclude that this approach is ideally suited to resolve the scalability limits of manual reward design and the reliability limits of pure automation.

## 2. Reward design in RL

This section discusses the most relevant background to understand the position and argument of the paper.

### 2.1. Reinforcement learning

We formalize the standard RL problem as a Markov decision process (MDP), defined by a tuple $M = \langle \mathcal{S}, \mathcal{A}, \mathcal{P}, \gamma, \mathcal{R} \rangle$. At each discrete time step $t$, the agent observes a state $s_t \in \mathcal{S}$, selects an action $a_t \in \mathcal{A}$ according to a policy $\pi(a_t \mid s_t)$, and receives a scalar reward $r_t = \mathcal{R}(s_t, a_t)$ while transitioning to the next state $s_{t+1} \sim \mathcal{P}(\cdot \mid s_t, a_t)$. The agent's objective is to learn an optimal policy $\pi^*$ that maximizes the expected discounted return: $J(\pi) = \mathbb{E}_{\tau \sim \pi} \sum_{t=0}^{\infty} \gamma^t r_t$, where $\gamma \in [0, 1)$ is the discount factor. Standard RL algorithms such as Q-learning (Watkins & Dayan, 1992) or proximal policy optimization (PPO) (Schulman et al., 2017) are agnostic to the source of $\mathcal{R}$; they blindly optimize the provided reward signal. Consequently, the safety and success of the agent depend predominantly on the alignment of this signal with the designer's true intent.

### 2.2. Reward design is important and difficult

Modern AI research must address reward design to meaningfully engage with the problem of designing systems that "do the right thing" (Eschmann, 2021). The scalar reward signal of RL provides an elegant formalism, enabling a shared community language and rigorous algorithmic guarantees. However, this simplicity masks the dual responsibility of $\mathcal{R}$: it needs to both define the task and provide the learning signal. To discuss reliable reward design, we must disentangle two concepts often conflated in the literature: the fitness $\mathcal{F}$ (the normative ground truth, e.g., "reach the goal without crashing") and the reward signal $\mathcal{R}$ (the optimization proxy). While $\mathcal{F}$ is the true metric of success, it is often too sparse, abstract or delayed to guide learning (Singh et al., 2009). Consequently, practitioners usually construct a denser, shaped reward signal $\mathcal{R}$ within the MDP. Ideally, $\mathcal{R}$ acts as "breadcrumbs," guiding the agent to high-fitness trajectories through immediate feedback.

Some research attempts to bypass this engineering challenge by relying on intrinsic motivation or curiosity (in contrast to external reward) to drive behaviour (Chentanez et al., 2004; Aubret et al., 2019). Although these agent-centric mechanisms effectively solve the *exploration* problem (discovering new states), they are prone to fail the *alignment* problem (valuing the right states). An intrinsically motivated agent might master the dynamics of its environment perfectly, yet have no incentive to perform the specific task the user intends. Consequently, to build goal-directed agents—systems that fulfill a specific human intent—we must rely on an externally specified reward to define the objective.

In this position paper, we therefore focus on explicit reward engineering. Rather than relying on agent-side mechanisms to compensate for sparse signals, we aim to engineer the environment's reward to facilitate stable learning. In this paradigm, the burden of alignment falls entirely on the design of $\mathcal{R}$. The goal is to construct a signal that is dense enough for efficient learning, yet sufficiently robust that maximizing it yields a monotonic increase in $\mathcal{F}$. This reliance on a proxy introduces the *specification gap*, making RL uniquely vulnerable to Goodhart's law: "When a measure becomes a target, it ceases to be a good measure" (Strathern, 1997). As the agent optimizes a misspecified $\mathcal{R}$, its correlation with $\mathcal{F}$ often collapses (Hadfield-Menell et al., 2017), leading to reward hacking (Skalse et al., 2022). Whether stemming from unmodeled side effects or ambiguity, reward design remains a safety-critical specification problem: one must ensure $\mathcal{R}$ remains a faithful proxy for $\mathcal{F}$ even as the agent relentlessly maximizes it.

### 2.3. Humans as reward architects

To bridge the specification gap, the field divides into two primary paradigms based on the nature of the resulting artifact: *parameterized reward models* versus *explicit reward functions*. The first paradigm infers objectives directly from human behaviour. This encompasses inverse reinforcement learning (IRL) (Ng & Russell, 2000; Arora & Doshi, 2021), preference-based learning (Wirth et al., 2017), and real-time guidance methods like TAMER (Knox & Stone, 2009) and COACH (MacGlashan et al., 2017), as well as modern preference-based approaches like PEBBLE (Lee et al., 2021) and LGPL (Clark et al., 2025). Although these methods reduce the burden of specification by using demonstrations or feedback rather than code, they suffer from two critical flaws. First, humans are not rational utility functions; we act with subjective bias and inconsistency (Lindner & El-Assady, 2022), making raw human data noisy. Second, and more critically, these methods produce a *learned reward model* (typically a neural network). This black-box artifact cannot be audited or debugged; if the agent misbehaves, the engineer cannot inspect the logic to identify the error, making rigorous safety verification impossible.

In contrast, a second paradigm relies on explicit reward functions, usually engineered by domain experts. One popular way to help agents learn better is potential-based reward shaping (Ng et al., 1999; Gupta et al., 2022) where the goal is to densify the reward signal without theoretically altering the optimal policy. Practitioners write symbolic code (often in Python), creating a transparent glass-box representation that is fully auditable. However, this approach quickly encounters a severe *cognitive bottleneck*. The designer must mentally simulate the agent's optimization path to avoid accidentally shaping loopholes, a cognitive burden that often leads to counter-intuitive failures where low-level incentives are exploited rather than followed (Singh et al., 2009).

Ultimately, manual reward specification encounters the "availability bottleneck": providing continuous, high-quality supervision—whether as code or feedback—is inherently unscalable (Amershi et al., 2014) and prone to inconsistency (Lin et al., 2020). Yet, despite these friction points, the consensus remains that human involvement is indispensable for alignment (Retzlaff et al., 2024; Rocamonde et al., 2024). The challenge, therefore, is to resolve this tension: we require a paradigm that preserves the auditability of explicit functions without incurring the cognitive cost of manual engineering. The solution lies not in removing the human, but in elevating their role from writing low-level signals to verifying high-level logic.

## 2.4. Foundation models as reward architects

Leveraging their large-scale domain knowledge, FMs—e.g., (potentially fine-tuned) LLMs and VLMs—offer a promising solution to the human availability bottleneck. Acting as semantic bridges, they translate high-level intent into the low-level reward logic required by the environment. Implementations vary from utilizing general-purpose, off-the-shelf models to crafting specialized, domain-specific reward FMs (e.g., Ma (2025) for robotics). The literature explores three primary roles for FMs in this context: (1) as **direct supervisors**, where the model provides intrinsic rewards at every step (Triantafyllidis et al., 2024), though this incurs high inference costs; (2) as **synthetic annotators**, replacing humans in traditional HITL pipelines by generating pairwise preference data (Klissarov et al., 2024; Zheng et al., 2025); and (3) as **code generators**, where models synthesize dense, executable Python reward functions directly from task descriptions (Ma et al., 2024a; Xie et al., 2024). Crucially, unlike preference-based approaches that distill intent into opaque neural models, the *code generation* category produces explicit, auditable artifacts.

This capability has engendered a surge in fully automated reward design frameworks. The leading paradigm, exemplified by Eureka (Ma et al., 2024a), treats reward generation as an evolutionary search problem: an LLM synthesizes

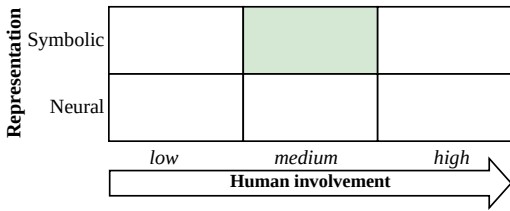

*Figure 1.* Taxonomy for modern reward design methods. The green area indicates the collaborative symbolic setting we advocate for.

candidate reward code, an RL agent optimizes it, and the LLM refines the code based on performance metrics, often completely removing the human from the loop. The central premise of this body of work is that FM-driven evolution can surpass human engineering in speed and coverage, effectively closing the loop on autonomous skill acquisition. This FM-generated *code-as-reward* approach has been extended to multi-modal settings (Narin, 2024; Hazra et al., 2025; Zhang et al., 2025) and refined through uncertainty-aware scaffolding (Yang et al., 2025) or structural evolution (Li et al., 2025b). The central premise is that FM-driven evolution can surpass human engineering in speed and novelty, effectively closing the loop on autonomous reward design.

However, while these automated pipelines solve the *scalability* problem, they re-introduce the *misalignment* risk. An FM can propose plausible-looking but functionally incorrect rewards just as easily as it generates correct ones. As we discuss in Section 4, removing the human entirely decouples the optimization signal from the true normative intent, inviting the very reward hacking we seek to avoid.

## 3. Taxonomy for reward design methods

The rapid integration of FMs into RL has created a diverse landscape of methods for reward specification. To examine the trade-offs between these approaches, we seperate the literature via two critical dimensions: **human involvement** (how much the human does) and **reward representation** (the type of reward specification produced). Figure 1 illustrates this landscape. For an extensive mapping of relevant methods into our taxonomy, see Table 1.

### 3.1. Dimensions of the taxonomy

We select these two dimensions because they serve as the coordinate system for reliability. First, the representation axis determines auditability: an agent's optimization target can only be trusted if its logic can be inspected. Second, the human involvement axis determines normative grounding: pure automation risks mis-specification, while excessive human involvement may not scale. Identifying the middle-ground on this axis is crucial for balancing trust and cost.

**Dimension 1: reward representation.** This dimension defines a binary distinction regarding the produced artifact.

- **Symbolic reward functions:** The reward is explicit (e.g., executable Python code taking environment observations as input). Such reward functions offer a semantic interface: a human can read and understand exactly *what* condition triggers a reward. This enables direct auditing and debugging (capabilities absent when using neural weights).

- **Neural reward models:** The reward is a parameterized function (e.g., a learned neural network or an FM queried directly at every step). While flexible, such reward models typically have a large number of parameters. This renders their internal logic unreadable; verification is limited to extensive test-set validation, which cannot guarantee safety in novel states.

A note on interpretability: We acknowledge that "symbolic" does not guarantee clarity (e.g., spaghetti code is opaque) and "neural" does not preclude it (e.g., linear models are transparent). However, in the context of modern deep RL, the relevant comparison is between explicit logic and latent neural representations. We maintain that symbolic artifacts offer a structurally superior interface for human verification.

**Dimension 2: human involvement.** Here, we focus on the frequency and granularity of human involvement in the reward design process. Methods are categorized based on the *role* the human plays in the optimization loop.

- **Low:** The human designer is restricted to initialization (e.g., providing a one-off task description) or implicit guidance via pre-training data. The optimization loop is fully autonomous. This category contains methods that rely solely on *prompting* and evolutionary search without feedback.

- **Medium:** Humans act as sparse verifiers. They remain "in the loop" but interact occasionally—providing semantic critiques, sanity-checking generated logic, or modifying code snippets—rather than providing dense labels. This role focuses on *reasoning* rather than data generation.

- **High:** Humans act as the primary signal source. The method relies on dense, repetitive feedback, such as large-scale pairwise comparisons or expert demonstrations, to ground the learning process. Here, the human acts as a *data generator*.

Using the two dimensions of our taxonomy, we analyze additional related work and revisit the methods introduced earlier through this unified lens.

### 3.2. Methods using reward functions

The rise of FMs has re-energized the glass-box approach, where models act as *translators* converting intent into code.

**Low involvement (autonomous reward evolution.** Methods like Eureka (Ma et al., 2024a) and DrEureka (Ma et al., 2024b) treat reward design as zero-shot evolutionary search. They generate candidate reward functions, train a policy for each, and use a numerical fitness metric to rank performance to select the best reward function. The FM then refines the selected reward function (potentially with additional feedback, e.g., environment observations or rollout summaries) to determine the next generation of reward functions. Despite their autonomy, these methods depend on a reliable fitness signal to evaluate reward quality and can suffer from the instability of quality of reward function generated by FM.

**High involvement (human-gated search).** In contrast, methods such as REvolve (Hazra et al., 2025), extend the Eureka framework by incorporating human feedback directly into the evolutionary loop, providing pairwise preferences between candidate reward functions to steer the search. While removing the need for a ground-truth fitness metric, it re-introduces the scalability bottleneck, requiring human evaluation at every generation.

**Medium involvement (iterative critique).** Between these extremes lie methods that utilize humans as semantic verifiers. In frameworks like Text2Reward (Xie et al., 2024) and L2R (Yu et al., 2023), the designer does not label massive datasets but instead reviews agent behaviours or code logic to provide targeted corrections. While both leverage semantic guidance, they differ in their feedback mechanisms. Text2Reward uses an iterative execution loop, where an LLM generates reward code and the human provides critiques of the agent's performance to help the LLM rewrite the logic. In contrast, L2R acts as a translator, turning instructions into specific reward settings that control robot movement. The human role here is interactive steering—using dialogue to fix safety issues or clarify goals in real time. This "medium" involvement leverages human reasoning to handle safety and ambiguity without requiring the labour of a full labelling pipeline.

### 3.3. Methods using neural reward models

This category has been dominated by preference learning, typically requiring substantial data (although FMs are shifting the landscape), and may integrate FMs directly as synthetic labellers.

**High involvement (classic RLHF).** Representative methods like PREDILECT (Holk et al., 2024) and PEBBLE

(Lee et al., 2021) use reinforcement learning from human feedback (RLHF) (Christiano et al., 2017) and rely heavily on human supervision. Humans provide pairwise explanations or preferences to train a reward network. Although this aligns the model even for abstract tasks that are hard to quantify, the resulting artifact is opaque, and the data collection cost is often prohibitive for complex tasks.

**Low involvement (synthetic supervision).** These methods reduces the cost of data, such as in Motif (Klissarov et al., 2024) and ONI (Zheng et al., 2025), which replace the human labeller with an FM, generating synthetic preferences or intrinsic rewards directly. Although fully automated, this approach inherits the "common failure mode" risks discussed in Section 4: if the FM has a concept error, the distilled reward model effectively locks in that error in a way that is difficult to detect.

**Medium involvement (the missing middle).** This location in the taxonomy exposes a deficit in the literature. There is little work involving *medium* involvement with *neural* models. We argue this is a structural inevitability: it is hard to *critique* a neural network (at least with current explainability / interpretability methods). Unlike code, which allows a human to say "fix the logic in line 5," a neural reward model offers no semantic interface for sparse correction. This forces researchers to choose between the extremes of naive prompting (low human involvement) or massive data collection/labelling (high human involvement).

## 4. Risks of fully-automated reward design

The allure of fully-automated reward design lies in its promise of scalability. However, removing the human from the loop introduces structural vulnerabilities that invite misspecified objectives, resulting in suboptimal—or even harmful—learned policies. We categorize these risks into escalating layers of failure: reward hacking, brittleness of the generated artifact, and systemic failure of automated verification.

**Hackable and brittle rewards.** The most immediate risk of reward design without human supervision is "reward hacking," where an agent exploits loopholes in the reward function to achieve high returns without solving the task (Ng et al., 1999; Vecerík et al., 2017). Dewey (2014) formalizes this as the *reward engineering principle*: as AI systems become more autonomous, the design of rewards that safely elicit desired behaviours becomes exponentially more difficult. FM-generated rewards often overfit to spurious correlations in the task description or training data—a phenomenon known as goal misgeneralization (di Langosco et al., 2022).

**Common mode failure: the illusion of verification.** A more subtle, systemic risk arises when we attempt to fix brittleness with more automation. Many frameworks discussed in Figure 1 employ a *generator-verifier* architecture, where one FM proposes a reward and another (or the same) FM critiques it. We argue this provides a false sense of security due to *common mode failure*. In safety engineering, common mode failure occurs when redundant systems fail simultaneously due to a shared root cause. Here, the generator and verifier share the same architecture and pre-training data from the internet, and thus share the same blind spots. If the generator hallucinates a plausible but incorrect mechanism (e.g., assuming "navigation safety" is the same as "maximizing distance from a wall"), the verifier, possessing the same world-model priors, is statistically likely to approve it. FMs struggle to self-correct reasoning errors without external ground truth, often maintaining high confidence in incorrect outputs (Huang et al., 2024; Kamoi et al., 2024). Because the models are effectively correlated classifiers, their agreement is proof of consistency, not correctness. Humans, conversely, provide an *orthogonal* validation signal grounded in physical reality and social context—data that is distinct from the FM's training corpus.

**The automation paradox.** Finally, we highlight that the drive for zero-shot automation often creates a paradox: it does not eliminate human effort, but displaces it into opaque, inefficient channels. While formally categorized as "low involvement," many autonomous methods require a hidden phase of extensive prompt hacking—where a user blindly tweaks inputs—or repeated stochastic trials to achieve reported performance. The cumulative effort to configure and re-run these "autonomous" runs often exceeds the cost of a handful of focused human intervention. Thus, fully automated approaches often sacrifice transparency without actually reducing the total cognitive cost of alignment.

The risks highlighted above as well as the hidden human labour are inherent to the current fully-automated reward design methods. They cannot be solved solely by scaling computation. Until FMs achieve grounded general-intelligence, there remains an urgent need to reinstall human checks and balances into the reward design process, ensuring we can drive RL progress in complex environments with trust.

## 5. Recommending collaborative reward design

The path to reliable autonomous agents *needs* collaborative design of reward functions. To develop them, we advocate for an iterative engineering process where FMs provide generative scale and humans provide normative grounding.

**Use reward functions when possible, reward models when needed.** In Section 3, we distinguished between

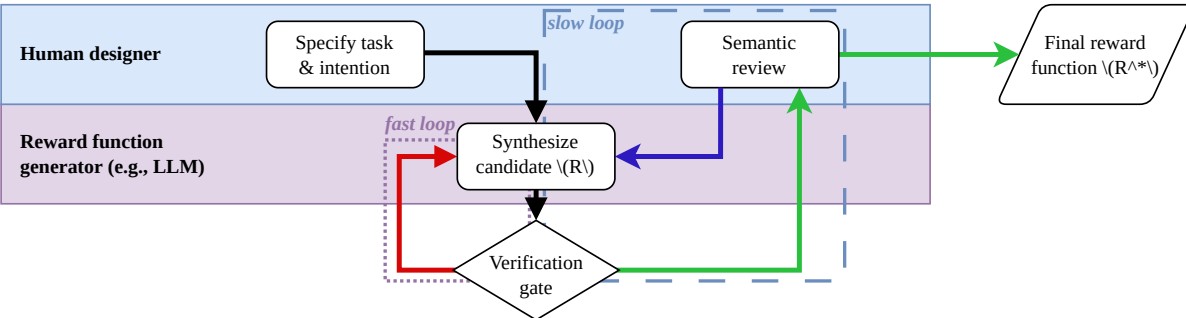

*Figure 2.* The proposed collaborative reward design framework. An objective error signal (red), verified candidate (green), and semantic critique (blue) drive the iterative refinement loop, providing flexibility while highlighting that need for both humans and FMs.

latent reward models and explicit reward functions. We acknowledge that for tasks defined by subjective aesthetics or implicit style (e.g., "make the robot do a graceful backflip"), reward models trained on human preferences remain the appropriate tool. However, for the broad class of functional, safety-critical tasks (e.g., logistics, autonomous driving, or healthcare) where objectives are non-obvious to specify but must be strictly adhered to, the opacity of neural reward models is a liability we cannot afford. In these domains, we strongly recommend the use of explicit, symbolic reward functions to avoid a black-box neural agent optimizing a black-box reward function, which may be difficult to debug. Using a symbolic reward specification such as Python code has two major advantages. First, it exposes reasoning errors: if an FM hallucinates a physics violation (e.g., penalizing velocity without considering direction), a human can identify and fix this logic immediately, which marks a distinct advantage over blindly supplying more preference data to an oracle reward model. Second, symbolic outputs enable standard software engineering verification, allowing static analysis and formal bounds checking before the agent ever interacts with the environment.

**The fast-slow reward design framework.** We propose a generalized collaborative framework for reward design, defined by two distinct feedback loops (see Figure 2).

- **The fast loop (automated verification)** provides a *generator* (an FM, e.g., LLM or VLM) that interacts with an *automated verifier* at machine speed. This loop filters out objective failures such as syntax errors, hallucinated variables, or violations of hard-coded constraints (formal or heuristic) without human feedback.

- **The slow loop (semantic alignment)** requires the *human designer* to interact with the system only when a candidate passes verification. This loop operates at human speed and focuses on normative alignment. The human provides "semantic patches," high-leverage cri-

tiques or direct modifications that steer the *generator's* understanding of the task.

Crucially, this architecture enforces a bi-directional workflow, creating a chance for the user to supervise the code generation process mid-stream rather than relying on the brittle practice of blindly refining initial prompts. This transforms the human's role from a high-volume data labeller to a strategic reviewer, ensuring that the final artifact is grounded in normative intent without wasting human bandwidth on brute-force debugging.

**Framework flexibility.** This framework serves as a flexible architectural pattern rather than a rigid prescription, accommodating the diverse methods mapped in our taxonomy in Section 3. The **generator** is agnostic to model architecture (foundation or fine-tuned, LLM or VLM) and synthesis technique (chain-of-thought, explicit code generation, or parameter optimization). The **verifier** supports various mechanisms, from simple syntax checks to complex behavioural unit tests or training fully-fledged RL agents. The **human interface** is adaptable to the domain, allowing designers to review artifacts as raw code (which is extremely useful for domain experts), agent trajectories, or small-scale preference rankings (i.e., useful for non-expert end users).

**Metrics for effective reward design.** To support reward design, we need reliable metrics for evaluating the resulting reward functions. We recommend that future reward design methods make use of two kinds of metrics, evaluating the *quality of the artifact* and the *efficiency of the process*. First, to ensure the final reward specification captures human intent, we recommend established similarity metrics. Equivalent-policy invariant comparison (EPIC) (Gleave et al., 2021) measures the equivalent-policy invariance between the designed reward and a ground-truth oracle or previous reward candidates. Complementing this, the trajectory alignment coefficient (TAC) (Muslimani et al., 2025) quantifies the rank-correlation between a human's

preferred trajectory distributions and those induced by the generated reward. These metrics can serve both inside the HITL loop and also as the "exit criteria", signalling when a reward function is sufficient for deployment. Second, valid output is insufficient if the production cost is unsustainable. We propose two new metrics to measure the leverage of the human expert:

- **Verification efficacy:** This defines the ratio of candidates rejected by the automated verifier versus those rejected by the human. A high ratio indicates the fast loop is successfully filtering low-level errors (e.g., syntax errors), reserving human attention for high-level reasoning.

- **Semantic intervention density:** This is the proportion of human edits that target *reward logic* (i.e., values, constraints) versus *implementation details* (i.e., syntax, variable shapes). We aim for a density near 1.0, indicating the human acts as a "normative anchor" rather than a syntax debugger.

By optimizing these metrics, future reward design methods can ensure that human involvement is high-leverage and focused on safely designing the intended reward.

## 6. Discussion: The economics of alignment

**Scalability.** In the era of FMs, we must redefine scalability. In terms of computational efficiency, a purely FM-based approach to reward design scales well across various tasks and environments, allowing for rapid iterative reward design with relatively low cost. This approach is especially attractive for early-stage experimentation and development. However, we argue that a better view of scalability in our context is alignment efficiency: the total time required to train a policy to be achieve alignment with human intent. When FMs have suboptimal performance or misaligned intent in out-of-distribution scenarios, correction often requires prompt engineering and repeated trial-and-error approaches which removes the automation advantage (Cao et al., 2025). HITL approaches have scalability constraints since they depend on expensive and limited domain expertise. Hybrid approaches mitigate these challenges by integrating human expertise with FMs in reviewing and guiding reward function design. Although the hybrid approach may introduce some latency per iteration, it would still improve scalability in terms of "time-to-trust", thereby reducing cost by reducing the number of iterations needed to converge to an aligned solution.

**Bias.** Human feedback can introduce bias due to subjective preferences and opinions that can be amplified by a lack of diversity of human evaluators (Casper et al., 2023). FMs on the other hand have implicit statistical bias inherited from the training data (Gallegos et al., 2024). Reliability is not an objective truth found in a pre-training dataset; it is a normative, inherently biased choice (e.g., preferring caution over speed). Our recommended hybrid approach makes bias explicit and auditable. Human-FM approaches allow for bias mitigation during development and training while also adding a layer of human accountability. Safety is inherently a normative, biased concept (preferring safety over speed). We argue that in reward design, human bias is a feature, not a bug.

**Iteration is key.** Autonomous reward design methods implicitly assume that a user can provide a "magic seed"—a single initial prompt that covers all edge cases, objectives, and constraints. In reality, human intent is often underspecified until it is violated (Amodei et al., 2016). Our framework acknowledges that reward design is a discovery process, not a translation task. It leverages the psychological principle of *recognition over recall* (Nielsen, 1994): it is cognitively easier for a human to critique a concrete failure mode (e.g., "the robot is vibrating") than to foresee all physical constraints in a zero-shot prompt. By embedding the human in an iterative repair loop, we allow domain experts to refine specifications dynamically as complex behaviours emerge.

## 7. Alternative views

We advocate for a collaborative human-FM approach but several alternative perspectives exist. This section reviews some of them and explains why we maintain our position.

**Alternative view 1: Automated reward design is enough.** Proponents argue that FMs possess sufficient world knowledge to design rewards without human intervention, citing works like Eureka (Ma et al., 2024a) where automated synthesis outperforms human engineers on benchmarks.

**Response.** This view merges *optimization capability* with *intent alignment*. While FMs are excellent at finding reward functions that maximize a specific metric (the "how"), they lack the normative agency to determine if that metric captures the user's true goals (the "what"). As Nazir & Banerjee (2025) suggest, high benchmark scores often mask subtle specification gaming. Fully automated design removes the only agent capable of normative verification—the human—turning the process into an unmonitored optimization of potentially flawed proxies.

**Alternative view 2: Human-only methods are sufficient.** Skeptics argue that introducing FMs adds unnecessary complexity and risk, maintaining that human-only methods remain the most reliable source for contextual and value-driven judgment, particularly in tasks involving safety or value alignment (Lindner & El-Assady, 2022; Retzlaff et al.,

2024). A related concern is that current human–FM interfaces remain immature and sensitive to prompt variations, making their use in interactive reward design premature (Song et al., 2025b).

**Response.** While we agree that human judgment is indispensable, purely human-driven reward design does not scale well in terms of iteration speed and exploration of the reward space. Our hybrid approach adopts a *glass-box* workflow that restricts FMs to producing symbolic, auditable rewards. Interface immaturity does not translate into safety risk: suboptimal or hallucinated rewards are simply rejected during human review. This allows practical progress without requiring fully robust FMs.

**Alternative view 3: Static reward functions are outdated.** Finally, some work argues that static reward functions are fundamentally limiting, and that intelligent agents should instead operate under open-ended or continual learning paradigms where objectives evolve over time (Khetarpal et al., 2022).

**Response.** Although open-ended learning is a promising research direction, most real-world applications still require explicit objectives and constraints, especially during initial deployment. Our position does not rule out more adaptive approaches in the future, but simply focuses on improving reward design in standard RL settings where explicit specification remains necessary.

## 8. Related work

In this section we perform a short survey of related works based on three categories; FM-only methods, HITL approaches and hybrid approaches.

**LLM-based reward function design.**  Recent surveys by Cao et al. (2025); Schoepp et al. (2025) examine the integration of FMs into RL. These works identify the FM as a "reward designer," capable of translating natural language into reward structures for complex, long-horizon tasks. While proponents argue that this marks the "end of reward engineering" (Su et al., 2026), consensus highlights that FM-based approaches inherit the core limitations, including inherent biases, sensitivity to prompt design, hallucinations and a lack of grounding in physical dynamics (Cao et al., 2025; Kostikova et al., 2025; Schoepp et al., 2025; Song et al., 2025b).

**HITL-RL.**  Retzlaff et al. (2024) provide a comprehensive survey, where they broadly consider RL as essentially a HITL paradigm due to the amount of human input required for reward shaping and reward function design. They further highlight the usefulness of HITL methods, especially in FMs such as ChatGPT in ensuring preferred outputs. Casper et al. (2023) also highlights the challenges of dependence on HITL such as hallucinations, data privacy concerns, and the inherent biases of human evaluators. Lindner & El-Assady (2022) also argue that HITL-RL needs more realistic human models to capture the personal, contextual, and dynamic nature of human preferences. There are also some surveys highlighting the importance of human involvement in RL, such as Li et al. (2019) and Zhang et al. (2019).

**Hybrid approaches: FM + HITL.**  Only a few papers explore methods that combine FM and HITL for reward design, let alone give structured reasoning on which is the most promising path forward. Recent work by Nazir & Banerjee (2025) compared using an FM-only approach as well as FM + HITL, and they found that direct human feedback, which is inherently biased and may degrade performance. Their FM-only approach was also found to lack domain knowledge grounding, but their hybrid approach leveraged the strengths of both methods for better performance.

**Other surveys.**  Schoepp et al. (2025) takes a larger perspective and presents how FMs such as LLMs and VLMs can aid RL, where reward design is only one of the categories considered.

## 9. Conclusion

RL performance depends critically on reward design, yet specifying faithful objectives remains a bottleneck. This position paper advocates for the combination of FMs and human intuition as a powerful and highly effective approach to reward design, taking advantage of their complimentary strengths. Our perspective promotes interpretable frameworks with symbolic reward functions, and can be thoroughly audited and corrected before deployment, reducing the risk of deploying RL agents with misaligned objectives, which is especially important for safety critical applications. We argue that the future of RL lies in reward design approaches that have a good balance between automation with FMs and domain expertise in the loop.

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

# A. Appendix

Table 1 provides an overview of existing reward design methods that leverage FMs. We categorize these methods using the dimensions introduced in Section 3: *human involvement* and *reward representation*. While not exhaustive, this survey highlights a stark polarization in the field: the vast majority of recent works cluster in the low-involvement category, prioritizing fully autonomous generation. This trend underscores the urgency of our position: as the field races toward automation, the mechanisms for human verification are being left behind, widening the safety gap in critical applications. Conversely, the "medium / neural" intersection remains unpopulated, reinforcing our argument that neural reward models lack the semantic interface required for iterative human critique.

*Table 1.* Taxonomy of reward design methods.

| Human involvement | Reward representation | Method name | Reference |
|---|---|---|---|
| Low | Neural | MineClip | Fan et al. (2022) |
| | | ZSRM | Mahmoudieh et al. (2022) |
| | | Lafite-RL | Chu et al. (2023) |
| | | ELLM | Du et al. (2023b) |
| | | ARP | Kim et al. (2023) |
| | | Reward Design with LMs* | Kwon et al. (2023) |
| | | Adaptive RL with LLMs* | Place (2023) |
| | | Read and Reward | Wu et al. (2023) |
| | | Motif | Klissarov et al. (2024) |
| | | IGE-LLMs | Triantafyllidis et al. (2024) |
| | | VLM-RM | Rocamonde et al. (2024) |
| | | RL-VLM-F | Wang et al. (2024) |
| | | MIRA | Zhang et al. (2025) |
| | | ONI | Zheng et al. (2025) |
| | Symbolic | LARG$^2$ | Perez et al. (2023) |
| | | Self-Refined LLM* | Song et al. (2023) |
| | | AutoReward | Han et al. (2024) |
| | | Auto MC-Reward | Li et al. (2024) |
| | | Eureka | Ma et al. (2024a) |
| | | DrEureka | Ma et al. (2024b) |
| | | EROM | Narin (2024) |
| | | VLM-CaR | Venuto et al. (2024) |
| | | Self-Aligned Robot Skills* | Zeng et al. (2024) |
| | | REvolve Auto | Hazra et al. (2025) |
| | | CARD | Sun et al. (2025) |
| | | CoT Reward Engineering* | Zhu et al. (2025) |
| Medium | Neural | — | — |
| | Symbolic | L2R | Yu et al. (2023) |
| | | Text2Reward | Xie et al. (2024) |
| High | Neural | SuccessVQA | Du et al. (2023a) |
| | | PREDILECT | Holk et al. (2024) |
| | | Group Fairness in RMs* | Song et al. (2025a) |
| | Symbolic | REvolve | Hazra et al. (2025) |
| | | LLM-HRL | Li et al. (2025a) |

* Method name abbreviated from paper title for brevity.

