# OpenReview forum: "Position: Reliable Reward Design Needs Both Humans and Foundation Models"
_ICML.cc/2026/Position_Paper_Track — Submitted to ICML 2026 Position Paper Track_

### Official Review · Reviewer_6mg2 · 2026-03-12

**Significance:** 3
**Argument Clarity:** 2
**Rating:** 4
**Confidence:** 3

**Questions:**

These questions mostly follow up on the weakness points.

1. Could the authors provide more concrete examples of how the proposed framework would be applied in practice? For instance, are there specific tasks, experimental setups, or benchmarks where this human–FM collaborative reward design pipeline could be evaluated under different tasks?

2. Do the authors have thoughts on how this framework could be adapted to these different domains, and whether different types of tasks might require different forms of human involvement or verification? In a realistic development loop, how would the interaction between humans and foundation models look in practice?

**Alternative Views Section:**

Yes

**Compliance With Llm Reviewing Policy A Conservative:**

Affirmed.

**Discussion Potential:**

2

**Final Justification:**

The authors have addressed my concerns in the rebuttal. I would keep a positive view on this paper.

**Paper Summary:**

The paper focuses on the problem of reward design in RL, arguing that specifying accurate rewards with both human and foundation models is one of the main bottlenecks for deploying RL systems in complex environments.

As the position paper, the core argument is that neither humans nor foundation models alone are sufficient for reliable reward design. Foundation models can help scale reward design by translating task descriptions into executable reward functions, but fully automated pipelines risk introducing subtle errors or misalignment. Human experts provide normative grounding but cannot scale to large numbers of tasks. The main framework-wise contribution is a collaborative framework where foundation models generate candidate rewards in a fast automated loop, while humans act as a slower verification layer to review and refine them, aiming to balance scalability and reliability.

Overall, I think the paper positions this human–FM collaboration as a practical path toward more reliable reward design in RL.

**Position:**

Yes

**Position In Title:**

Yes

**Related Work:**

3

**Strengths And Weaknesses:**

**Strengths**

1. [position and significance] The position addressed in this paper is significance and timely. First RL is increasingly relevant across many domains, including robotics, LLMs, and reasoning models. Within RL, reward design remains a core challenge, as the behavior of the agent fundamentally depends on how the reward is specified. From this perspective, the paper targets a highly meaningful problem.

2. [context and related work]  The paper provides a clear and well-formulated overview of existing approaches to reward design. The discussion of different paradigms, including manual reward engineering, neural reward representations, and the use of foundation models for reward generation is well organized and helps clarify the current landscape.

3. [discussion potential] The topic is likely to stimulate discussion within the community. For researchers working on RL, alignment, or LLM-based agents, the paper offers a useful conceptual summary of current directions and trade-offs in reward design.

**Weaknesses**

1. I felt the framework is still a bit too absract. While the proposed collaborative framework (e.g., Fig. 2) is intuitive, the overall idea of combining automated generation with human verification is also a fairly common paradigm. It would be better to include more concrete analysis or examples. For instance, (even if a full technical formulation is not necessary for a position paper), the authors could discuss more concretely how reward optimization might happen in practice, how the two sources (humans and FMs) interact in the loop, or provide illustrative case studies, experimental setups, or potential benchmarks.

2. Connection to concrete tasks and broader communities. Related to the above point, the discussion could benefit from grounding the ideas in more specific application settings. For example, the paper could connect the framework more explicitly to existing paradigms such as RLHF or RL-from-AI-feedback in LLMs, or reward design in robotics and embodied RL tasks. Using a few concrete task examples might make the proposed position more tangible and easier to be reachable by different communities that have common needs for reward design.

**Support:**

2

---

> ### Author Rebuttal · Authors · 2026-03-31
>
> We thank the reviewer for the constructive feedback. We appreciate the request for concrete applications and clarification of how the interaction loop adapts to the unique requirements of different task domains.
>
> ---
>
> **Question 1: Concrete examples of practical applications, experimental setups, and benchmark evaluation.**
>
> While our framework is deliberately abstract to serve as a broad design paradigm, we have expanded Section 5 with the following details:
>
> * **Illustrative example of our framework "in action"**: Consider a household robot, tasked with clearing fragile glassware. A human prompts the foundation model (FM) to generate Python code maximizing speed and minimizing distance to the bin (fast loop). An automated verifier checks for syntax. However, during simulation, the agent learns to violently throw the glass into the bin. Observing this physical exploit, the human issues a semantic patch (slow loop): "Add a strict penalty for high object velocities." The FM updates the logic, and the agent learns smooth transportation.
> * **Example of framework implementations**: L2R and Text2Reward are examples of reward design methods that match our vision of human-FM collaboration (see Table 1). We have also identified another such work [1], which we have added to the medium-human-involvement / symbolic-representation category. This illustrates the range of suitable tasks, from robotic manipulation and locomotion (Text2Reward, L2R) to autonomous car racing [1]. Furthermore, our framework addresses the pressing need for actionable intermediate rewards in the physical sciences [2].
> * **Evaluation & benchmarks**: While benchmarks like B-Pref (for preference-based RL) and RewardBench (for evaluating FM reward models) strategically evaluate alignment, they are unsuitable for evaluating programmatic reward functions. We propose evaluating on standard benchmarks (e.g., Meta-World, RLBench) using the specific metrics defined in Section 5: verification efficacy (does the FM catch its own syntax errors?) and semantic intervention density (is the human acting as a normative anchor rather than a syntax debugger?). Additionally, the trajectory alignment coefficient (Muslimani et al., 2025) can quantify alignment with human intent.
>
> ---
>
> **Question 2: Adapting the framework to different domains and the practical realities of human-FM interaction.**
>
> While our framework is domain-agnostic and intended as a broad architectural paradigm, it is likely that different types of tasks will require different concrete implementations:
>
> * **Ideal candidates**: The framework adds most value in safety-critical, constraint-heavy, and physically grounded tasks, making it vital for the robotics and embodied AI communities. Here, the sim-to-real gap makes the opacity of neural models a severe liability. While deep RL policies process high-dimensional perception neurally, the rewards used to train them can often be designed symbolically. For example, prominent legged robotics work [3] uses symbolic rewards structured around nine distinct, manually tuned components to ensure physical safety (e.g., penalizing foot slip or joint velocity). Black-box neural models cannot currently guarantee such rigid constraint adherence, making collaborative symbolic design essential.
> * **Poor candidates**: Our framework is not intended to replace existing methods for highly subjective, aesthetic, or conversational tasks (e.g., aligning an LLM's values). For these, explicit reward code is unrealistic, and traditional RLHF (neural representation / high human involvement via preference labeling) remains the correct tool.
>
> ---
>
> **Weakness 1: Addressing the novelty of the proposed framework.**
>
> While combining automated generation with human verification might seem like a natural progression, it is not the current trajectory of the field—as noted in our response to reviewer E4H3, the community overwhelmingly favours fully autonomous pipelines. We do not claim a single monolithic algorithm that "solves" reward design. Rather, we propose a necessary architectural paradigm shift. We find the community currently moving from one extreme (unscalable manual human engineering) to another (unsafe, fully autonomous FM generation). Our framework argues that future research must improve the communication channels and verification gates between FMs and domain experts. Our goal is to strengthen the vocabulary for the RL community to stop asking "How do we remove the human?" and start asking "How do we maximize the human's normative leverage?"
>
> ---
>
> We incorporated these clarifications into the paper and appreciate the help grounding our framework.
>
> ---
>
> [1] Ma, M., et al. Automated reward design for Gran Turismo. NeurIPS, 2025.
>
> [2] Kalinin, S. V., et al. Bringing ML to the real world: Rewards are all we need. APL Machine Learning, 4(1), 2026.
>
> [3] Rudin, N., et al. Learning to walk in minutes using massively parallel deep reinforcement learning. CoRL, 2022.

---

> > ### Author Rebuttal · Reviewer_6mg2 · 2026-04-03
> >
> > Thank you for the thoughtful and detailed response, especially the discussion of Q1 with concrete examples. I would keep my positive rating. However, I would still choose (c) here, since I still feel that the main point is largely aligned with what is already considered common sense in the RL community (about reward design). So while I would not say the paper lacks novelty entirely, I am not yet convinced that it offers a sufficiently new perspective to warrant being framed as a position paper.

---

### Official Review · Reviewer_E4H3 · 2026-03-12

**Significance:** 4
**Argument Clarity:** 3
**Rating:** 5
**Confidence:** 4

**Questions:**

- When reward designs become sufficiently complex, to what degree will humans be able to keep up and safely/reliably audit these designs?
- A primary alternative, as outlined in the paper, is that automated design is enough. That current models lack the normative agency to determine if the metric captures the underlying goals, but humans can. While this is a resonable response, I question (1) to what variance humans exhibit normative agency relative to LLMs and (2) in the limit of LLM intelligence, whether this will hold true in a few years time.
- On symbolic functions in incresing complexity, how do the authors propose this is handled? As systems, particularly multi-agent systems, become incresingly complex and inter-dependent, how can we retain reward designs that are tractable for humans to review?

**Alternative Views Section:**

Yes

**Compliance With Llm Reviewing Policy A Conservative:**

Affirmed.

**Discussion Potential:**

3

**Final Justification:**

The rebuttal addressed my concerns and I raise my score accordingly. This direction has relevance and importance in the growing world of agents and FMs and exploring ways to prevent reward hacking and the like are important.

**Paper Summary:**

The paper proses that reward design must be collaborative between humans and foundation models, balance scalability and reliability. The paper argues that closed validation loops, where foundation models design reward autonomously, are more of a translation task, rather than a grounding task. In following, humans should be added as a verification layer to ground these processes. The authors propose then that a "fast-slow" collaborative loop where systems have foundation models acting as automated generators and verifiers, paired with periodic human review. This is intended to maintain alignment and transparency, as well as safety in planning that models often overlook or hack around.

**Position:**

Yes

**Position In Title:**

Yes

**Related Work:**

4

**Strengths And Weaknesses:**

### Strengths
- The paper is well written and clear in its positioning
- The paper outlines related work in the context of its position well, helping readers ground their understanding. The outlined taxonomy is clear and easy to follow and outlines the position well.
- The proposed fast-slow loop is straightforward, and its positioning is well supported.
- The paper can serve good discussion in the ICML community.
- The position is broadly applicable to many fields of ML.

### Weaknesses
- While the position is well thought out, I do not believe it is particularly in contention in the broader community.
- While the critique of fully automated reward designs seems plausible, the paper lacks clear evidence of this. While not explicitly required for a position paper, this would have strengthed the position.
- In incresingly complex real-world problems, it is not clear than symbolic functions will be more auditable than neural reward models.

**Support:**

3

---

> ### Author Rebuttal · Authors · 2026-03-31
>
> We thank the reviewer for the opportunity to clarify the boundaries of human oversight and the utility of symbolic reward functions.
>
> ---
>
> **Question 1: Human cognitive bandwidth in increasingly complex reward designs.**
>
> We agree that as task and reward complexity grow, human cognitive bandwidth becomes a bottleneck. However, this calls for evolving the human's role, not eliminating it. We view the second question’s premise—that foundation models (FMs) will become more capable over time—as a core strength of our framework rather than a vulnerability. The FM’s role can naturally expand from generation to translation and critique. Rather than forcing humans to manually audit an overwhelmingly long reward function, our framework benefits from the general evolution of FMs, employing them as sophisticated alignment assistants. The FM can generate semantic diffs of code changes, summarize algorithmic intent, and proactively surface edge-case trajectories for review. By trusting FMs to handle dense technical translation, we empower humans to scale their oversight and focus on high-level normative approval. Whether humans can indefinitely maintain oversight over increasingly autonomous AI is an active debate [1], but our framework provides an immediate step forward by utilizing the FM to preserve human cognitive bandwidth. Regarding viability with symbolic reward functions, see our response to Q3.
>
> ---
>
> **Question 2: Variance of human normative agency vs. future LLM intelligence.**
>
> While FMs will become increasingly capable technical assistants, we must distinguish between normative capability (the ability to mimic or predict moral reasoning) and normative authority (the legitimacy to dictate values).
>
> * **Regarding variance**: While FMs can approximate human consensus in well-defined scenarios [2], this alignment diverges on contested objectives—where reward design is most sensitive. The relatively high variance of humans in moral judgment reflects genuine deliberation over competing values, whereas FM consistency reflects distributional averaging that often collapses when disagreement and nuance matter most.
> * **Regarding the limit of FM intelligence**: Deciding whose values a reward function serves is a normative choice, not a technical optimization problem. Greater intelligence does not confer authority: Humans are the ground truth for human values by definition. Regardless of how capable FMs become, our framework retains humans in the loop because they uniquely hold the normative authority and deployment liabilities behind a reward specification.
>
> ---
>
> **Question 3: Tractability of complex symbolic reward functions.**
>
> The core advantage of symbolic representations expressed as code is that they support established software engineering principles like modularity, abstraction, and unit testing. As RL systems—including multi-agent systems, as mentioned in the question—increase in complexity, monolithic reward equations indeed become intractable. However, symbolic functions can be factored into independent programmatic sub-components (e.g., `calculate_collision_penalty()` and `calculate_team_cohesion_score()`). In our framework, the FM in the fast loop independently generates and unit-tests these modular sub-components. The human in the slow loop then audits only the compositional logic (e.g., explicitly adjusting the weights between the collision penalty and the team score). Just as programmatic representations have been successfully used to generate interpretable and verifiable agent policies [3], this symbolic decomposition of rewards provides a foundation for auditability that is structurally impossible with black-box neural models, directly mitigating vulnerabilities like model misspecification and reward misgeneralization [4].
>
> ---
>
> **Weakness 1: Position contention in the broader community.**
>
> We argue that while human-centered AI is an agreed-upon ideal, it is not the current research reality in reward design. Our taxonomy in Table 1 reveals a stark automation bias: 26 out of 33 recent methods advocate for fully autonomous pipelines. The community is prioritizing FM scalability at the expense of human normative grounding. Our paper provides a necessary intervention to rebalance this trajectory, framing human oversight as a structural necessity rather than a post-hoc check.
>
> ---
>
> We have incorporated these clarifications into our paper and hope they further demonstrate the value of our contribution to the community.
>
> ---
>
> [1] Holzinger, A., et al. Is human oversight to AI systems still possible? New Biotechnology, 2025.
>
> [2] Garcia, B., et al. The moral Turing test: Evaluating human-LLM alignment in moral decision-making. arXiv:2410.07304, 2024.
>
> [3] Verma, A., et al. Programmatically interpretable reinforcement learning. ICML, 2018.
>
> [4] Pitis, S. Failure modes of learning reward models for LLMs and other sequence models. ICML Workshop, 2023.

---

> > ### Author Rebuttal · Reviewer_E4H3 · 2026-04-03
> >
> > I remain borderline in acceptance of the work. My primary concern remains around human cognitive bandwidth and increasing FM intelligence, and how this framework ultimately falls into the same problem of a fully neural reward. Namely, by using FMs to translate/summarize the intent, write unit tests, code, etc., the FM is equally positioned to obscure its true goals/intent/reward from the user. [1] Is a very interesting example of such a phenomenom. We are effectively just moving the black box one layer up. The human in the loop can be fed a slightly decoupled summary of what is actually going on. Anecdotally, current FMs are very good at confirming human-in-the-loop biases, and I don't see this going away.
> >
> > [1] Anatomy of a Reward Hack: A Real Story from the Latest GPU Mode NVFP4 Competition
> >  https://www.gpumode.com/news/reward-hacking-nvfp4

---

### Official Review · Reviewer_awJZ · 2026-03-13

**Significance:** 3
**Argument Clarity:** 2
**Rating:** 4
**Confidence:** 3

**Questions:**

See weakness

**Alternative Views Section:**

Yes

**Compliance With Llm Reviewing Policy A Conservative:**

Affirmed.

**Discussion Potential:**

3

**Final Justification:**

my concern is well resolved

**Paper Summary:**

This submission claims to outline a general area for improving reinforcement learning (RL) reward design through collaboration between humans and foundation models (FMs). The paper aims to consider a central concept: combining the scalability of FM-generated reward functions with human verification to improve reliability. It introduces a taxonomy of reward design methods based on human involvement and reward representation (symbolic vs. neural), and argues that a hybrid setting, symbolic rewards with moderate human oversight, is the most reliable. The paper proposes a fast–slow loop framework, where FMs generate and automatically verify reward code while humans provide high-level semantic corrections.

**Position:**

Yes

**Position In Title:**

Yes

**Related Work:**

3

**Strengths And Weaknesses:**

Strengths

1. Important and timely topic. The paper addresses reward specification, a key bottleneck in RL systems.

2. Clear conceptual taxonomy. The two-dimensional framework (human involvement × reward representation) helps organize existing approaches.


Weakness:

1. The paper assumes that humans can effectively identify and correct reward specification errors. However, reward hacking often emerges from subtle optimization dynamics that humans may not anticipate, and manual inspection may be slow or unreliable in complex environments.

2. The paper focuses mainly on reward specification while largely ignoring how RL optimization itself interacts with reward functions[1,2]. In practice, perfectly aligned reward functions are difficult to design, and policy learning can exploit even carefully engineered rewards. As a result, improving reward design alone may not sufficiently address reward misalignment.


[1] Gradient Regularization Prevents Reward Hacking in Reinforcement Learning from Human Feedback and Verifiable Rewards
[2] Correlated Proxies: A New Definition and Improved Mitigation for Reward Hacking

**Support:**

2

---

> ### Author Rebuttal · Authors · 2026-03-31
>
> We thank the reviewer for the insightful feedback and the opportunity to clarify our position.
>
> We agree that reward hacking is often driven by subtle optimization dynamics, and relying solely on human foresight is insufficient. We address both concerns below and have incorporated these points into our paper.
>
> ---
>
> **Weakness 1: The difficulty of manually anticipating subtle optimization dynamics and the risk of slow, unreliable human inspection in complex environments.**
>
> We agree that humans struggle to anticipate subtle optimization dynamics. As acknowledged in Section 2.3, manual reward engineering requires mentally simulating optimization paths to avoid loopholes, a cognitive burden that often causes failures. While we do not provide a magic bullet, our approach is designed to mitigate this limitation by providing a flexible structure that elevates the human's role from writing low-level signals to verifying high-level logic.
>
> As mentioned in our response to reviewer E4H3, our framework is highly flexible and the responsibility of how to surface issues lies with the authors of specific reward design methods. It accommodates naive approaches such as static code review, as well as intricate implementations that include full RL training loops or small policy rollouts to present the reward designer with common or edge-case trajectories.
>
> To specifically address the concern that manual inspection is slow and unreliable in complex environments, our framework provides structural mechanisms and metrics that shift the burden from human foresight to empirical review assisted by foundation models (FMs):
>
> * **Behavioural auditing**: By integrating environment rollouts into the slow loop, methods built on our framework can present designers with actual agent trajectories rather than forcing them to rely solely on foresight. When a subtle exploit emerges, the human observes it directly, leveraging the psychological principle of recognition over recall, as discussed in Section 6.
> * **Traceable logic**: Because our framework outputs explicit, symbolic reward functions, the human can trace observed behavioural exploits back to specific lines of logic and patch them immediately, significantly speeding up the inspection process.
> * **The FM as an alignment assistant**: FMs can proactively surface edge-case trajectories and summarize behavioural impacts, effectively scaling human oversight by translating complex dynamics into semantic questions.
> * **Monitoring human leverage**: To keep inspection sustainable, we propose monitoring metrics such as verification efficacy and semantic intervention density in Section 5. These serve as indicators to measure whether the automated fast loop is successfully filtering low-level syntax errors, helping to ensure the human's limited cognitive bandwidth is reserved for the high-level normative reasoning that requires true domain expertise.
>
> ---
>
> **Weakness 2: The limitations of addressing reward misalignment solely through specification, without optimization-level defenses.**
>
> We agree that improving reward design alone does not entirely solve misalignment. RL is uniquely vulnerable to Goodhart's law (see Section 2.2); agents will relentlessly attempt to maximize rewards, unbothered if the proxy's correlation with the true fitness collapses.
>
> Our paper intentionally isolates the reward specification problem, operating upstream of the optimization phase. Therefore, the highlighted algorithmic mitigations are perfectly complementary to our approach: Gradient regularization [1] and correlated proxies mitigation [2] introduce crucial method-side defenses against reward hacking. Building aligned RL agents ultimately requires tackling both sides of the problem:
>
> * **Reward specification**: Methods following our framework establish a reward function that serves as a semantically grounded proxy for the true intended goal. In complex environments, human common sense provides an orthogonal validation signal grounded in physical and social reality that FMs lack, ensuring behaviour aligns with human intent.
> * **Optimization**: Algorithmic regularizations [1, 2] act as the critical safety net, preventing the policy from violently overfitting to any remaining imperfections in the proxy.
>
> ---
>
> We appreciate this broader context and incorporated [1, 2] into our discussion, noting that collaborative reward design must be paired with robust RL optimization to mitigate reward hacking. We hope these revisions fully resolve the highlighted concerns while strengthening the paper's overall contribution.
>
> ---
>
> [1] Ackermann, J., et al. Gradient regularization prevents reward hacking in reinforcement learning from human feedback and verifiable rewards. arXiv:2602.18037, 2026.
>
> [2] Laidlaw, C., et al. Correlated proxies: A new definition and improved mitigation for reward hacking. ICLR, 2025.

---

> > ### Author Rebuttal · Reviewer_awJZ · 2026-04-04
> >
> > my concern has resolved, I will increase my score to 4

---

### Decision · Program_Chairs · 2026-04-30

**Decision:**

Reject

**Comment:**

This paper addresses a timely and important problem: as foundation models increasingly automate reward design in RL, the field lacks a principled framework for structuring human involvement. The taxonomy, the fast-slow loop framework, and the operational metrics (verification efficacy, semantic intervention density) are concrete contributions. All three reviewers maintained positive scores after the rebuttal, and the paper is well-written with strong related work coverage.

In terms of weaknesses, there is some feeling among reviewers that the position might already be obvious to the broader research community. It would strengthen the paper to include a discussion of conditions under which symbolic transparency might erode as system complexity increases.